# How EBV Infects: The Tropism and Underlying Molecular Mechanism for Viral Infection

**DOI:** 10.3390/v14112372

**Published:** 2022-10-27

**Authors:** Guo-Long Bu, Chu Xie, Yin-Feng Kang, Mu-Sheng Zeng, Cong Sun

**Affiliations:** 1State Key Laboratory of Oncology in South China, Collaborative Innovation Center for Cancer Medicine, Department of Experimental Research, Sun Yat-sen University Cancer Center, Sun Yat-sen University, Guangzhou 510060, China; 2Guangdong-Hong Kong Joint Laboratory for RNA Medicine, Guangzhou 510060, China

**Keywords:** Epstein–Barr virus (EBV), infection, tropism, disease, entry

## Abstract

The Epstein–Barr virus (EBV) is associated with a variety of human malignancies, including Burkitt’s lymphoma, Hodgkin’s disease, nasopharyngeal carcinoma and gastric cancers. EBV infection is crucial for the oncogenesis of its host cells. The prerequisite for the establishment of infection is the virus entry. Interactions of viral membrane glycoproteins and host membrane receptors play important roles in the process of virus entry into host cells. Current studies have shown that the main tropism for EBV are B cells and epithelial cells and that EBV is also found in the tumor cells derived from NK/T cells and leiomyosarcoma. However, the process of EBV infecting B cells and epithelial cells significantly differs, relying on heterogenous glycoprotein–receptor interactions. This review focuses on the tropism and molecular mechanism of EBV infection. We systematically summarize the key molecular events that mediate EBV cell tropism and its entry into target cells and provide a comprehensive overview.

## 1. Introduction

The Epstein–Barr virus (EBV) was the first identified human oncogenic virus, ubiquitously infecting over 90% of the world population [1,2]. As it is recognized as a class I oncogenic virus by the World Health Organization, EBV accounts for over 200,000 cases of cancer and 1.8% of all cancer deaths each year [3,4,5,6]. Additionally, EBV infection is associated with a series of autoimmune diseases, including multiple sclerosis (MS) and systemic lupus erythematosus (SLE) [7,8].

EBV is an enveloped virus belonging to family *Herpesviridae*. It contains a linear double-stranded DNA genome with a length of approximately 172K bp, encoding over 80 viral proteins and multiple noncoding RNAs [9,10,11]. EBV can infect multiple cell types but typically features a B cell–epithelial cell dual tropism [12]. The entry machine of the EBV is highly sophisticated, enabling its infection of different cell types [12,13]. As many as 13 glycoproteins are encoded by EBV, with eleven of them found on the virion envelope, and the entry process requires the coordination of multiple glycoproteins [14,15]. The core fusion apparatus glycoproteins gH/gL and gB are indispensable for viral entry among all herpes virus, while other glycoproteins play heterogenous roles in the viral tropism fate and viral entry process [13,16,17].

Hence, understanding how EBV establishes variable host tropisms and the underlying mechanism for EBV entry into different cell types could be the cornerstone for developing prophylactic vaccines and therapeutic agents against EBV infection in all kinds of susceptible cells. This review aims to summarize the present understanding of EBV cellular tropism, EBV-related diseases, and entry mechanisms to further point out the vital unanswered questions in this field.

## 2. EBV Tropism and Related Diseases

EBV has a characteristic latent-lytic lifecycle, during which EBV generally establishes latent infection in B cells and lytic infection in epithelial cells to generate progeny virions for transmission, and under certain stimulation, EBV in B cells could reactivate and enter lytic replication, with BZLF1 and BRLF1 as switch-driving genes [18,19,20] (Figure 1).

In this section, we would introduce EBV tropism and EBV-related diseases and mainly focus on the two major tropism cell types, B cells and epithelial cells.

### 2.1. B Cell

EBV was firstly identified in an endemic Burkitt lymphoma (BL) cell culture using electron microscopy [1]. Afterwards, the strong potential of EBV in transforming B cells was elucidated [21,22]. EBV established a lifelong latent persistence in B cells [23,24].

EBV is orally transmitted, and the saliva-derived EBV virions have been demonstrated with a B-cell-tropic glycoprotein composition, suggesting that B cells are the initial targets of EBV [25,26,27,28,29]. The lifelong persistence of EBV has been established in memory B cells, and two different models have been proposed to explain this process. The first supposed that EBV directly infects memory B cells, as was observed in patients with infectious mononucleosis (IM) [30,31]. The second model proposed that EBV initially infects resting naïve B cells residing in oropharyngeal lymphoid tissue and frives the infected B cells into a rapid expansion phase [32,33,34]. The infected B cells then enter the germinal center (GC) and come to a stage that viral genes are moderately expressed with viral latent membrane proteins LMP1 and LMP2 offering survival signals [35,36,37,38,39]. However, it is still unclear in this hypothesis how these cells finally differentiate into memory B cells. The EBV-infected memory B cells manifest the least viral protein expression and circulate in the peripheral blood [40,41]. Under certain stimulations, these EBV-infected memory B cells could reactivate, differentiate into plasma cells, and enter lytic replication [20,42,43] (Figure 1). Additionally, a recent study using single-cell transcriptomics to investigate the fates of B cells at the early stage of in vitro EBV infection and depict the dynamic host–virus interplay, as well as the resulting diverse fates of infected B cells. This study puts forward a model that EBV-infected B cells could differentiate into GC B cells, plasmablasts and early memory B cells that limitedly undergo GC reaction from the activated precursor state, dependent on the interplay between host cell and viral gene products [44].

The B-cell tropism accounts for a huge part in the pathogenesis of EBV. While the primary EBV infection is mostly asymptomatic during childhood, the delayed acquisition of primary infection in adolescents or young adults can lead to infectious mononucleosis, which features the EBV-infected B-cell proliferation and active T-cell response [45]. The occurrence of IM is associated with an increased risk in developing Hodgkin’s lymphoma (HL) and multiple sclerosis (MS) [46,47]. B-cell lymphomas, including BL, Hodgkin’s lymphoma (HL), diffuse large B-cell lymphoma (DLBCL), plasma blastic lymphoma, PEL, and so on, are found to be associated with EBV infection to varying degrees [48,49]. EBV infection could contribute to oncogenesis in multiple ways, in dependence of EBV-encoding RNA and viral proteins, such as EBV BART microRNAs promoting BL development through decreasing apoptosis-inducing protein expression [50,51,52].

### 2.2. Epithelial Cell

After the discovery of EBV in the BL cell line, scientists found an increased level of anti-EBV antibody titer in the plasma of nasopharyngeal carcinoma (NPC) patients [53]. It is an epithelial tumor prevalent in Southern China, Southeast Asia, North Africa, and the Arctic [54]. With a further study finding of the EBV genome in NPC tumor cells, the association between EBV infection and epithelial malignancy is gradually recognized and becomes a focus [55,56].

The current studies support that epithelial cells are a natural reservoir for EBV amplification and play important role in EBV transmission [57]. EBV lytic replication has been found in oropharyngeal cells during IM and in oral hairy leukoplakia, an epithelial hyperplasia that commonly occurs in patients coinfected with human immunodeficiency virus (HIV) [58,59,60]. Moreover, virions from saliva were found to harbor an epithelial cell-preferred composition of surface glycoproteins, which also gave a clue as to the idea that EBV from saliva are produced in epithelial cells [25] (Figure 1).

Now, a strong link between undifferentiated NPC and EBV infection has been elucidated, as the EBV genome presents in all NPC cells [61,62,63]. Additionally, approximately 10% of gastric carcinomas worldwide and certain types of salivary gland carcinomas are found pathologically associated with EBV infection [64,65,66]. Dissimilar to B cells, EBV infection in primary epithelial cells cannot directly induce cell proliferation or immortalization but results in growth arrest instead [61,67,68]. The current model supposes that pre-existent aberrant genetic or epigenetic alterations in preinvasive nasopharyngeal epithelium enable the establishment of EBV latent infection, which then confers survival benefits for the development of epithelium-origin malignancy [61,69,70,71,72].

### 2.3. NK/T Cell

Besides the typical B cell–epithelial cell dual tropism, EBV can also infect NK cells and T cells and is associated with the malignancy of these cell types, though in relatively rare cases [73].

In 1988, Jones et al. firstly identified EBV DNA in T-cell lymphoma [74]. Later, with the establishment of EBER in situ hybridization and other experimental methods, a variety of NK and T-cell lymphomas were found to be associated with EBV infection [75]. Moreover, EBV-positive NK or T-cell lines have been developed using patient specimens or through in vitro EBV infection of EBV-negative NK or T-cell lines, which further suggests that NK/T cells could be targets of EBV [76,77,78,79]. Studies have also revealed that, in tonsils and peripheral blood from IM patients, a small portion of EBV-infected T and NK cells was found. This evidence showed that the infection of NK/T cells could occur in the primary EBV infection [80].

### 2.4. Others

In leiomyosarcomas of transplant recipients or HIV-infected patients who have severely undermined T-cell immunity, the EBV genome could be detected, but EBV cannot be detected in leiomyosarcoma of immune-competent patients, suggesting that smooth muscle cells are an atypical target of EBV under the immune-compromised status of the host [81,82,83,84,85].

Instead of the direct infection of host cells to induce malignancies, EBV can also cause remote cytopathic effects. Recently, a large cohort study revealed that EBV infection was epidemiologically linked to multiple sclerosis (MS), an autoimmune neurodegenerative disease [86]. Another study found that EBV could contribute to the pathogenesis of MS through antigen mimicry between the viral protein EBV transcription factor EBV nuclear antigen 1 (EBNA1) and the host central nervous system protein glial cell adhesion molecule (GlialCAM) to induce autoimmune demyelination [87]. Besides MS, EBV infection is linked to a series of autoimmune diseases, including rheumatoid arthritis, Sjögren’s syndrome, and systemic lupus erythematosus, though the etiological role of EBV remains to be investigated [7,88,89,90,91]. These findings show that EBV could cause not only infected cells but remote uninfected cell cytopathy, expanding our current knowledge on the mechanism of EBV pathogenicity in human diseases.

Multiple cell types could be infected by EBV in vivo, and epidemiological and biological studies have demonstrated a variety of diseases related to EBV infection. It is now clear that EBV features a B cell–epithelial cell dual tropism, which is important to its latent-lytic lifecycle. However, it is currently unclear whether the infection of other cell types plays a role in EBV lifecycle and how EBV enters these cells. Additionally, as more and more evidence suggests that EBV could contribute to autoimmune diseases through cross-antigen mimicry, it is also worth investigating how EBV infection impacts uninfected cells and leads to remote cytopathy. More effort should be paid to elucidating the direct association between the EBV lifecycle and the transition between habitant cell types. These works could provide insight into EBV pathogenesis, which benefits the development of target therapies.

## 3. EBV Entry into Target Cells

Like other enveloped viruses, EBV adopts a similar viral entry strategy [92]. Surface viral glycoproteins on virion firstly bind attachment receptors to concentrate virions on the host cell surface to facilitate secondary receptor binding. Then, glycoproteins recognize entry receptors to trigger membrane fusion. Finally, the fusion protein adopts conformational changes to accomplish membrane fusion. However, unlike most enveloped viruses, such as SARS-CoV-2 relying on only one or two proteins to fulfill the viral infection process, EBV possesses and adopts a wide range of glycoproteins [13,93]. The function for each glycoprotein and collaboration of glycoprotein subgroups greatly differs in both entry processes (attachment, binding, and membrane fusion) and targeting infection cell (B cells, epithelial cells, etc.). As the entry machine of EBV is much more sophisticated, multiple receptors have been identified, participating in different infection processes and different cell type infections [94] (Table 1).

In this section, we summarize the current understandings on EBV entry into B cells and epithelial cells by systematically introducing the viral glycoproteins and host receptors involved in each entry process (Figure 2).

### 3.1. Attachment

Attachment is the process when the virus is concentrated upon the cell surface to facilitate further receptor binding. For example, a model is proposed that SARS-CoV-2 utilizes heparan sulfate for attaching to the host cell membrane and facilitating the binding of its entry receptor ACE2 [106]. EBV attachment to the host cell is mediated by several viral glycoproteins and host cell membrane proteins, tethering the virus to the cell membrane but not directly triggering viral entry [13,107]. There are differences between the mechanisms underlying the attachment of EBV to B cells and epithelial cells (Figure 2).

B cell. CD21/CD35, also known as the complement receptor type 2, complement receptor type 1 (CR2/CR1), which are mainly expressed on lymphocytes [108,109,110]. The interaction between EBV envelope glycoprotein, gp350/220 with the host CD21/CD35 is responsible for EBV attachment to B cells. As the most abundant glycoprotein on the envelope of EBV virions, gp350/220 binds CD21 and mediates the capping of CD21 and endocytosis of virions into B cells [109]. Soluble derivatives of gp350/220, as well as the soluble recombinants of CD21 were found able to inhibit the EBV infection of B cells when they were incubated with B cells or the viruses [111,112]. Although the binding of gp350/220 with CD21 is not strictly required for the EBV infection of B cells, gp350/220 absence reduced the B-cell infection efficiency [113].

Studies show that gp350/220 binds CD21 through its N-terminal 1-470aa, a glycan-free patch on its surface [95,112]. The predicted interaction region of gp350 and CD21 displays almost the opposite electronic potential, rendering a highly negatively charged surface on the glycol-free gp350 surface and positively charged area of the SCR1-SCR2 domain of CD21 ready for potential electrostatic interactions [112]. The surfaces of these two regions are also morphologically complementary. However, direct structural information of the protein complex is yet to come out to verify the exact interaction pattern.

For its crucial role in mediating B-cell attachment and its abundance on the virion surface, gp350/220 has been recognized as an ideal target for preventing EBV infection and EBV-related diseases [114,115,116]. Several monoclonal antibodies (mAb) targeting gp350/220 have been developed and identified to neutralize the EBV infection of B cells, and the well-characterized mouse origin mAb 72A1 has been further redesigned into a humanized form [117,118,119,120]. However, as the EBV vaccine antigen candidate that has received most attention before, gp350 immunization was shown to only confer partial protection against IM and no protection against asymptomatic EBV infection in a phase II clinical trial in young adults [113,121].

Epithelial cell. Unlike B cells, epithelial cells do not constitutively express CD21 or CD35 [122]. In a microdissection study of oropharyngeal epithelium, CD21 mRNA could only be detected in tonsils, suggesting that EBV may adopt different mechanisms in most epithelial cells to attach to the cell membrane during infection [123]. Moreover, the finding that antibodies targeting gp350/220 or CD21 failed to inhibit EBV virion attachment to cultured polarized oropharyngeal epithelial cells also supports this ratiocination [124].

BMRF2, an EBV glycoprotein with a RGD motif, was found to facilitate EBV attachment to epithelial cells through binding with β1, α5, and α3 integrins [124,125]. The RGD motif in the host extracellular matrix proteins interacts with cellular integrins to facilitate adhesion, and this motif is also found in other viral proteins and mediates viral attachment and entry [126]. Some studies found that antibodies against BMRF-2 and β1 integrin suppressed attachment and entry into epithelial cells but not B cells, and a lack of BMRF2 hindered EBV virions attachment to epithelial cells instead of B cells [124,125,127].

Otherwise, BMRF2 is found on the cell membrane of various EBV-infected cells, including lymphoblastoids, polarized oral epithelial cells, and hairy leukoplakia (HL) epithelium. Further, subcellular research revealed that BMRF2 was transported to the basolateral membrane of oral epithelial cells and colocalized with β1 integrins [125,128]. It further suggested that BMRF2 may also function in the cell-to-cell transmission of EBV. The EBV glycoprotein BDLF2 can interact with BMRF2 and function in cellular actin network rearrangement, through which BDLF2 coordinates with BMRF2 to facilitate cell-to-cell EBV infection by increasing cell membrane contacts via cytoskeleton manipulation [129].

EBV gH/gL was also found with a KGD motif, indicating a potential capability of the integrin-binding motif, while studies have found it could bind αvβ5, αvβ6, and αvβ8 integrin [130,131]. The interaction between EBV gH/gL and integrins is primarily considered to be a fusion-triggering event in epithelial cells, but a later study found that knocking out integrin αv did not influence the EBV fusion activity [132]. This result suggested that αvβ5, αvβ6, and αvβ8 integrins could be more likely attachment receptors rather than entry receptors.

### 3.2. Binding and Entry

Binding is a process of viral glycoproteins interacting with entry receptors, activating signaling cascades and triggering membrane fusion, which, finally, results in viral entry. Herpes virus binding and entry are generally mediated by multiple proteins. Among the glycoproteins, gH/gL takes the lead as the most critical apparatus inducing membrane fusion, while gB is the key molecule executing fusion process with the presence of gH/gL and other glycoproteins [13] (Figure 2).

B cell. After the initial attachment to the host membrane through gp350, EBV glycoprotein gp42 takes over to fulfill the next role in viral binding and entry. gp42 directly binds gH with its N-terminal region and, together with gL, forms gH/gL/gp42 heterotrimer. The gp42 in the heterotrimer could bind to the HLA class II molecule (HLA-II), a subtype of the histocompatibility complex mainly distributed on antigen-presenting cells (APC) such as B cells [133,134]. The interaction between gp42 and HLA-II results in the the conformational change of gp42 and is suspected to secondarily impact the conformation of gH/gL to triggers the pre- to post-conformational change of gB and achieve membrane fusion [100]. Studies found that virions in lack of gp42 can attach to B cells but fail to infect them, and host cell expressing CD21 but lacking HLA-II are resistant to EBV infection, which could be reversed by the rescue introduction of HLA-II [135,136]. Additionally, interrupting the interaction between gp42 and HLA-II with a soluble of form gp42, anti-gp42 mAb, or anti HLA-DR mAb could all inhibit the EBV infection of B cells [133]. This evidence all supports the vital role of gp42 in B-cell infection, and further structural studies provided us with more information.

In a study reporting the crystal structure of the gH/gL/gp42 heterodimer, the N-terminal of gp42 comprehensively binds gH domains II (D-II), III (D-III), and IV (D-IV), whereas its C-terminal shows limited binding with gH through the hydrophobic pocket (HP) interacting with the gH D-II domain [17]. Another structural investigation using negative staining electronic microscopy revealed that the complex had two different conformations: “open” and “close”, and the angle between HLA-DQ2 and gH/gL differed in the two statuses [137]. The number of the two conformation statuses was approximately even. Specifically, in the “close” status, two arms (gH/gL and HLA-DQ2) were closer to each other, which led to a shortened distance between their proximal membrane ends. This could play an important role in triggering the subsequent membrane fusion.

The characteristics of gp42 binding to HLA-II even displayed more influence on the viral tropism than its original function in binding and entry [135]. Due to HLA-II mainly expressing on B cells, the production of virion with the surface expression of virus glycoproteins, including gp42, could lead to in situ interactions of gp42 with HLA-II on the same cell membrane. A study found that B-cell-derived EBV displayed lower gp42 in comparison to epithelial cell-derived EBV and that a higher gp42 copy number on the virions could contribute to a higher infection efficiency in B cells and lower in epithelial cells, which further supported the in situ membrane interaction hypothesis and implied the potential switch role for gp42 in determining the viral tropism for EBV [138]. Based on this observation, EBV virions in saliva containing a high level of gp42 were supposed to be produced in epithelial cells [25].

Epithelial cells. In contrast to B-cell infection, the mechanism of EBV entry into the epithelial cells is more mysterious, and not until recent years did a series of research shed light on this field [14]. It is found that various receptors are involved in the binding and entry of EBV virions through variable binding to EBV gH/gL or gB.

Neuropilin-1 (NRP1) is a multifunctional protein and acts as a cofactor for cell entry in a variety of viruses [139]. It has been reported that NRP1 mainly interacts with proteins containing the carboxy (C)-terminal basic sequence motif (CendR motif), which contains a sequence of R/K/XXR/K such as SARS-CoV-2 spike [140]. Meanwhile, EBV gB possesses a highly similar and conserved R-X-K/R-R motif as a furin-cleavage site. Hence, it is hypothesized that NRP1 may interact with EBV gB through the CendR motif and play a role in EBV infection. A study demonstrated that gB could bind to NRP1 in vitro and the CendR motif deficiency would lead to an almost absent binding [141]. Further investigation showed that NRP1 knocked-out or soluble NRP1 pretreatment in the supernatant could inhibit the EBV infection of epithelial cells [141].

Non-muscle myosin heavy-chain IIA (NMHC-IIA) is an actin-binding protein, with properties of actin crosslinked and contractile regulated by phosphorylation [142]. Research identified the interaction between NMHC-IIA and gH/gL through the pull-down assay and mass spectrometry [143]. Although NMHC-IIA is normally located in the cytoplasm, there is evidence that NMHC-IIA is densely clustered on the surface of SLCs (sphere-like cells). A functional assay further suggested that the downregulation or blocking of NMHC-IIA reduced EBV infection in epithelial cells. NMHC-IIA interacted directly with EBV gH/gL in vitro. However, only cell surface NMHC-IIA was reported to enhance the EBV infection efficiency, whereas cytoplasmic NMHC-IIA did not, and the redistribution mechanism of NMHC-IIA was still unclear.

Ephrin type-A receptor 2 (EphA2) belongs to the largest receptor tyrosine kinase family and plays important roles in cell migration, proliferation, differentiation, and axonal guidance [144]. EphA2 has been found as a receptor for several other pathogens, such as Kaposi’s sarcoma herpesvirus (KSHV), a member of gamma-herpes virus [145,146]. Research found that it also plays an important role in EBV infection of epithelial cells through binding with EBV gH/gL [132,147]. It was further revealed that the overexpression of EphA2 promotes EBV infection in epithelial cells, while knocking out EphA2, or pretreatment with the soluble EphA2 protein, EphA2-blocking antibody, or EphA1 (natural ligand of EphA2), could significantly reduce epithelial cell infection. Moreover, previous studies found that EphA2 mediates KSHV endocytosis dependent on kinase activity, which was not required in EBV infection [146].

The recent structural analysis of EphA2 and EBV or KSHV gH/gL complexes showed that EBV gH/gL or KSHV gH/gL bound to the EphA2 channel and the surrounding region, and the N-terminal of EBV gL was inserted into the EphA2 channel. Compared with KSHV gH/gL, EBV gH/gL has a lower affinity to the receptor [99].

### 3.3. Membrane Fusion

The process of viral membrane fusion is unique to enveloped viruses. During virus entry, the viral and host cell membrane or intracellular endosome membrane approach contact and fuse with each other under the mediation of the membrane components, forming fusion pores and releasing viral nucleocapsid [148].

Different from other enveloped viruses, herpes virus membrane fusion is generally mediated by multiple glycoproteins and, finally, fulfilled by core fusion protein gB [94]. EBV gB is a type III fusion protein, featured by the coil-coil hairpin structure in post-fusion conformation [98]. With the previous studies of structurally homologous VSV fusion protein G and recently revealed prefusion structures of HSV-1 and HCMV gB, it is now commonly believed that the prefusion status of gB exists and may take major responsibility in triggering viral–host membrane fusion through structural arrangement for not only EBV but all herpes viruses [94,149]. However, plenty of research found that gB itself cannot initiate membrane fusion, and the minimal components for EBV to start up membrane fusion were gH/gL and gB, leading to a general hypothesis that gH/gL triggers the prefusion gB to initiate viral membrane fusion [13,150]. So that gH/gL could act as harbor for gp42 to bind HLA-II on B cells and main target binding to integrins, NMHC-IIA and EphA2 on epithelial cells, membrane fusion initiation could differ in either cell type (Figure 2).

B cell. The gH/gL/gp42 complex of EBV binds specifically to HLA-II, the B-cell receptor, promoting expansion of the hydrophobic pocket in the gp42 C-terminal region. This change decreases the angle between HLA-II and gH/gL, leading to closer transmembrane structures to each other, and is suspected to be critical for the follow-up membrane fusion [17,137]. The structure change induced by receptor binding may contribute to prefusion gB conformation change, prompt its metastable fusion status, and facilitate the completion of membrane fusion executed by gB [13].

Epithelial cell. Compared to B-cell infection, the binding and entry process are more versatile in the interaction pattern between the viral and host membrane during epithelial cell infection, leading to an even more complicated fusion machine [14]. As gH/gL could bind variable receptors on the epithelium, including EphA2 and NMHC-IIA, it may also act as a trigger for inducing gB-mediated membrane fusion during the receptor binding [94,132,143,147]. Studies have shown that EphA2 could significantly impact the membrane fusion induced by gH/gL and gB using the cell–cell fusion model. Moreover, the rescue of EphA2 could significantly retrieve the infection efficiency of EBV to epithelial cells, which indicates that EphA2 binding to gH/gL may be a start-up signal for membrane fusion. Moreover, instead of receiving external molecular stimulation from gH/gL, gB could also be directly influenced by receptor binding [132,147]. As mentioned above, gB could bind to NRP1 with its CendR motif. Further evidence also showed that the overexpression of NRP1 could elevate the fusion activity of gH/gL and gB, indicating the potential role of NRP1 in the direct triggering of gB for membrane fusion [141]. However, due to that, the absence of gH/gL could completely invalidate the fusion activity of gB, whether the direct signal from gB binding to the receptor is critical for gB-mediated fusion remains unknown.

Due to the complexity of the viral membrane component, the entry process of EBV into host cells manifests a heterogenous behavior on both the tropism level and receptor level. Despite shared entry procedures, including attachment, binding, and membrane fusion, EBV adopts almost divergent sets of glycoprotein–host receptor interactions to accomplish the viral infection, which brings challenges for depicting the detailed molecular mechanism for EBV infection and difficulties for precise vaccine or viral inhibitor development. Compared to the clearer tropism of EBV, the exact process for how EBV completes these processes in single-protein resolution remains to be revealed. Further effort for EBV study in identifying new receptors, resolving the prefusion status of gB, finding the exact molecular interaction between gH/gL and gB, and elucidating the gB conformational change mechanism would be significant for accurate drug and vaccine development for broadly inhibiting EBV infection in all cell types.

## 4. Discussion

EBV is one of the most widely transmitted viruses in the world, infecting more than 90% of the population and being associated with a variety of autoimmune diseases and malignancies [6]. Despite huge healthcare burdens caused, there is currently no effective antiviral agent or vaccine for EBV to inhibit its infection [14]. EBV can establish variable host tropisms, leading to a heterogenous virus cycle and viral gene expression pattern [18]. In the meantime, the molecular machine of viral glycoproteins recognized different receptors on different host cell types, further complicating the investigation of the exact mechanism of host selection and detailed process of viral infection. Hence, deeper knowledge of tropism establishment and the viral entry process are significant for the generation of a potent inhibitor or prophylactic vaccine for EBV.

The discovery of the EBV genome or related expressed genes in different malignancies shaped the early awareness of EBV infection in different cell types. Through in-depth investigation of the association between lymphoid malignancies such as Burkitt’s lymphoma (BL) or epithelial malignancies such as nasopharyngeal carcinoma (NPC) and EBV, more information was obtained to gradually reveal the connection between the viral habitant after primary infection, virus lifecycle for latency and lytic amplification, and tropism transition between cell types, leading to the conclusion that integrative protection is required for establishing a sterile immunity against EBV infection due to the B cell–epithelial cell cycle and preference of the latency status in B cells of EBV. The recent findings on the EBV EBNA1-derived peptide inducing multiple sclerosis (MS) through the mimicry of GlialCAM further suggest that EBV tropism did not limit its capability of causing cytopathy in remote uninfected cells [87]. However, how EBV tropism is decided and what molecular machine realizes the tropism switch would be the underlying question for the accurate development of antiviral agents for EBV.

The advancement of biological and structural approaches in studying viral infection behavior provides more chances for investigating EBV infection at a single-protein resolution. In similar strategies, researchers found the entry process of EBV into host cells could also be divided into attachment, binding, entry, and membrane fusion, during which more and more evidence showed a strong connection between distinct glycoprotein–receptor interaction pattern and host cell type [13]. This phenomenon suggests an underlying logic for the molecular machine of viral glycoproteins mediating viral infection preference and behavior in the confrontation of host cells, especially B cells and epithelial cells. EBV gp350 or gp42 is found mainly participating in B-cell attachment or binding. In comparison, gH/gL and gB are found broadly acting during attachment, binding, and membrane fusion for both B cells and epithelial cells, making them currently more immune-focused for neutralizing antibody or vaccine development. Moreover, the gp42 level on EBV virion is found connected with the host cell preference due to the role in B-cell binding and potential in situ neutralization of HLA-II during B-cell lytic replication [135]. All these findings indicate that the host tropism could be significantly influenced by both the host cell receptor expression pattern and viral glycoproteins assigned during infection and support the idea that deeper investigation of the complete fusion machine composition, acting process, and their structural information would be fundamental for both understanding the detailed molecular mechanism of viral infection and the accurate design of vaccines and drugs against broad EBV infection.

## Figures and Tables

**Figure 1 viruses-14-02372-f001:**
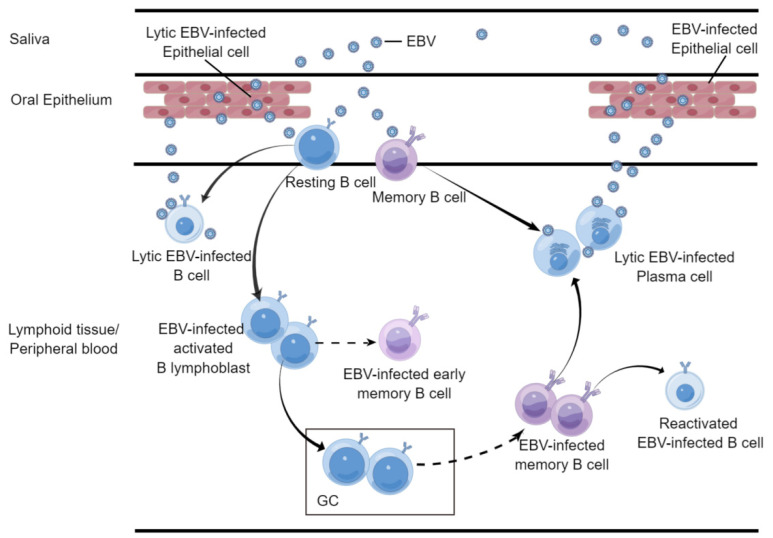
Model illustrating the infection of EBV in human. EBV is usually transmitted through saliva and first comes across oropharynx and tonsils of the new host, where it primarily infects naïve B cells and possibly memory B cells and epithelial cells as well. EBV establishes latent infection in most naïve B cells, while a small fraction of them might enter productive lytic infection. The latently infected naïve B cells are activated and undergo a phase of rapid expansion as a result of EBV latent gene expression. These activated infected B cells then enter the germinal center (GC), transiting to a more restricted form of viral latency. Finally, the infected B cells differentiate into resting memory B cells, where EBV nearly completely stops its gene expression and maintains its lifelong persistence, but the details of this process are still poorly understood. The EBV-infected resting memory B cells recirculate in the peripheral blood. Under certain stimulations, these resting memory B cells can differentiate into plasma cells, reenter a lytic infection state, and produce infectious virions. These virions lack in gp42 and are more epithelium-tropic. After infecting epithelial cells, the virus replicates lyses the cells, releasing the B-cell-tropic virions with high levels of gp42 into the saliva and starting a new round of transmission. Figdraw (www.figdraw.com) was used to generate this figure.

**Figure 2 viruses-14-02372-f002:**
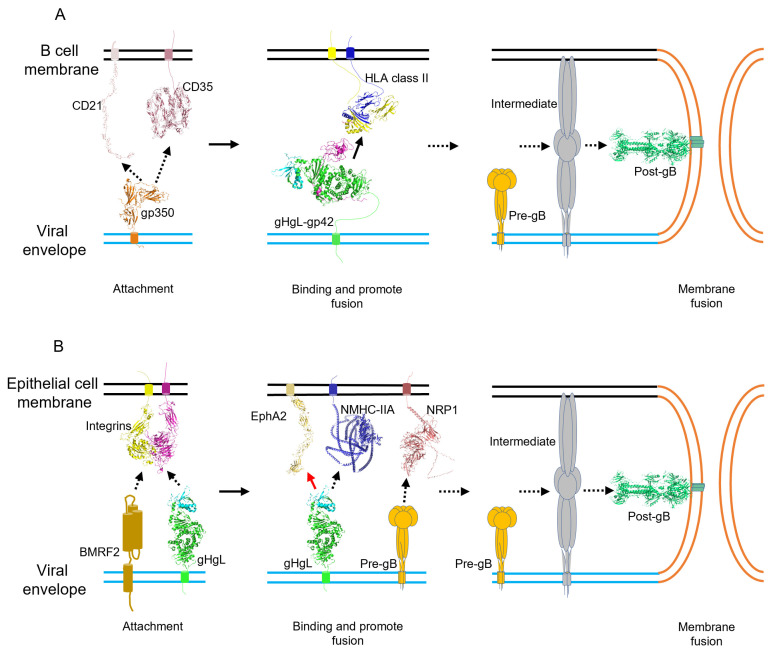
The process of EBV entry into host cells. (**A**) Entry of EBV into B cells through endocytosis. Firstly, EBV gp350 attachment with CD21 or CD35, and tethering EBV to B-cell membranes. Then, gH/gL-gp42 binds to receptor HLA class II. This interaction lets gH/gL-gp42 enable interacting with the prefusion form of gB. The interaction leads to a series of structural changes in gB and ultimately fusion of the viral membrane with the cell membrane. Extended rearrangements within gB (intermediate) enables insertion of the fusion loops into the B-cell membrane and the refolding of gB forms post-fusion gB, which mediates the merging of the two membranes. (**B**) The entry of EBV via direct fusion with epithelial cell membranes. Firstly, gp350/220 interacts with CD21 and/or BMRF2/gH/gL, with integrins tethering EBV to the epithelial cell membrane. After this, gH/gL binds to EphA2/NMHC-IIA or gB interacts with NRP1, producing a membrane fusion signal. The signal is passed to gB and causes a series of changes in its configuration. Extended rearrangements within gB trigger insertion of the fusion loops into the epithelial cell membrane, followed by the refolding of gB to post-fusion form and mediates the merging of the two bilayers. The structures are shown as cartoons: structures of CD21 (2GSX) [97], EBV gp350 (2H6O) [95], the complex of gH/gL (3PHF) [103], the complex of gp42-HLA-DR1 (1KG0) [101], post-fusion gB (3FVC) [98], and EphA2 (2X10) [104] and modified using the PyMOL Molecular Graphics System, Version 2.1.0 Schrödinger, LLC. The structure of integrin alpha5beta1 (3VI4) [105] was used as a representative model of integrins interacting with EBV membrane proteins. The structure files were retrieved from the PDB database. The structure of NMHC-IIA, NRP1, and full-length CD35 are generated by AlphaFold Protein Structure Database (https://alphafold.ebi.ac.uk/).

**Table 1 viruses-14-02372-t001:** Receptors/binding factors for EBV membrane proteins/glycoproteins.

Viral Protein	Receptor/Binding Factor	Target Cell	Function	Structure
**Membrane protein**				
BMRF2	Integrins	Epithelial cell	Viral attachment	None available complex structure
**Glycoproteins**				
gp350	CD21	B lymphocyte	Viral attachment	gp350 ecto-domain (PDB:2H6O) [95] CD21 SCR1-SCR2 (PDB: 1LY2) [96] CD21 (PDB: 2GSX) [97] None available complex structure
	CD35	B lymphocyte	Viral attachment	None available complex structure
gB	NRP1	Epithelial cell	Viral binding and promote membrane fusion	Post-fusion gB extracellular domain (PDB: 3FVC) [98] None available pre-gB structure None available complex structure
	EphA2	Epithelial cell	Viral binding and promote membrane fusion	None available complex structure
BMRF2	Integrins	Epithelial cell	Viral attachment	None available complex structure
gHgL	Integrins	Epithelial cell	Viral attachment	None available complex structure
	NMHC-IIA	Epithelial cell	Viral binding and promote membrane fusion	None available complex structure
	EphA2	Epithelial cell	Viral binding and promote membrane fusion	gHgL-EphA2(Ligand binding domain) (PDB: 7CZE) [99]
gHgL-gp42	HLAII	B lymphocyte	Viral binding and promote membrane fusion	gp42 (PDB: 3FD4) [100] gp42-HLA-DR1 (PDB: 1KG0) [101] gHgL-CL40-gp42_N-domain (PDB: 5W0K) [102] gHgL-gp42-E1D1 (PDB: 5T1D) [17]

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
