# Peer review of "How EBV Infects: The Tropism and Underlying Molecular Mechanism for Viral Infection"

_viruses, 2022, doi:10.3390/v14112372_

Round 1
Reviewer 1 Report
The manuscript by Bu et al. is a short and well structured review addressing the current knowledge on Epstein-Barr virus (EBV) tropism and entry mechanisms. This is an overall well-written review that however needs some attention for English and typographic errors. Overall, the manuscript provides an interesting review that will be of interest for the field as a review of the knowledge and still-to-be-discovered mechanisms that explain EBV infection.
- B cells and epithelial cells should in moist of the cases by written in the plural form rather than B cell or epithelial cell.
-SARS-CoV-2 on line 182 is not well spelled.
- Fig. 2: 'intermidiate' is not English, should be "intermediate". The usage of 3D structure representations together with drawings of the main domans of viral glycoproteins is strange and it is difficult to understand what the authors excatly want to show in the last part of the figure representing pre-gB, intermediate and post-gB. It should be clarified in the figure.
Author Response
Dear reviewer:
Thanks for your comments concerning our manuscript entitled “How EBV infects-the tropism and underlying molecular mechanism for viral infection” (Manuscript ID: viruses-1985118). We have conscientiously modified the corresponding parts. The modified parts have been marked in red font in the manuscript, the revision instructions can be seen in the attachment.

Reviewer 2 Report
The review by Bu et al is a solid review of the molecular mechanisms by which EBV glycoproteins mediate entry into the two major cell types EBV infects B cells and epithelial cells. The authors take into consideration the literature and present the information in a clear and precise manner. This is a really nice review.
Minor Comments
The introduction and section 2 need to be checked for grammar extensively as a number of grammatical errors were present. It is especially apparent when compared to the other sections of the review. It comes across as two different people wrote different part. Particular examples are below with plenty more in there. A thorough check for the entire review would be a good idea.
Line 32: Use of could is weird
Line 74: The start of the sentence is weird
Line 76: B cell should be plural
Line 78: Sentence structure is weird and tropical is an odd word choice over tropic
Line 84: Typo
Line 86: Transit is a weird word choice
Line 87: Typo
Author Response

(The authors gave the same response as above.)
